# Vitamin K1 and K2 in the Diet of Patients in the Long Term after Kidney Transplantation

**DOI:** 10.3390/nu14235070

**Published:** 2022-11-29

**Authors:** Małgorzata Kluch, Patrycja Bednarkiewicz, Magdalena Orzechowska, Piotr Grzelak, Ilona Kurnatowska

**Affiliations:** 1Department of Diagnostic Imaging, Polish Mother’s Memorial Hospital Research Institute, 93-338 Lodz, Poland; 2Department of Internal Medicine and Transplant Nephrology, Medical University of Lodz, 90-153 Lodz, Poland; 3Department of Molecular Carcinogenesis, Medical University of Lodz, 90-752 Lodz, Poland

**Keywords:** kidney transplantation, vitamin K, diet

## Abstract

Vitamin K, especially its K2 form, is considered to be a protective factor against developing vascular changes and bone lesions that are common complications in kidney transplant (KTx) recipients. There is a growing number of studies showing that KTx patients are at risk of vitamin K deficiency. The aim of this study was to evaluate the intake of vitamin K1 and K2 in the diet of patients in the late period after KTx. During a routine visit at one outpatient transplantation clinic in Central Europe, a diet survey questionnaire was filled in by 151 clinically stable KTx recipients and compared with medical history, anthropometric measurements and laboratory tests. Mean vitamin K1 intake was 120.9 ± 49 μg/day and vitamin K2 (MK, menaquinone) intake 28.69 ± 11.36 μg/day, including: MK-4: 25.9 ± 9.9 μg/day; MK-5: 0.1 ± 0.2 μg/day; MK-6: 0.2 ± 0.4 μg/day; MK-7: 0.2 ± 0.23 μg/day; MK-8: 1 ± 1.9 μg/day; MK-9: 0.9 ± 2.3 μg/day; and MK-10: 0.2 ± 0.5 μg/day. Our study showed that KTx recipients’ diets contained adequate amounts of vitamin K1, whereas the intake of vitamin K2 seemed insufficient.

## 1. Introduction

Cardiovascular (CV) diseases and bone disorders (next to malignancies and infections) are the most common complications in kidney transplant (KTx) recipients [1,2,3,4]. Vitamin K is considered to play a protective role in both the CV and skeletal systems [5,6,7]. The vitamin exists in two biologically active forms: K1 (PK, phylloquinone) and K2 (MK, menaquinone). MK is classified from MK-4 to MK-13 depending on the number of attached isoprenoid groups [8]. The main dietary source of PK is green vegetables [5], while MK is primarily produced by bacteria and is found in high concentrations in fermented foods, such as pickled vegetables and cheeses. Fair amounts of vitamin K2 can be found in meat, egg yolk andbutter but also in fermented soybeans—a Japanese delicacy called “natto” [5,9]. Both forms of vitamin K are cofactors for the enzyme γ-glutamylocarboxylase, which activates proteins involved in hemostasis, bone metabolism, vascular wall cell metabolism, cell growth and apoptosis [5,10]. Vitamin K1 has been attributed to a greater role in blood coagulation [11], while K2 is a cofactor for matrix Gla protein (MGP) carboxylation [5,12]. There are strong suggestions that the active form of MGP is an inhibitor of vascular calcification [13,14] and may inhibit artery calcification, e.g., through binding calcium and phosphorus ions and preventing their deposition in the artery walls [15]. It was shown that an elevated level of dephosphorylated uncarboxylated MGP (dp-ucMGP, a marker of vitamin K2 insufficiency) is connected with greater arterial wall calcification in the general population as well as in patients in all stages of chronic kidney disease (CKD) [6,16]. A higher level of dp-ucMGP also correlates with the presence of atherosclerosis and coronary artery calcification in patients with advanced stages of CKD [17]. It was shown that most KTx patients have vitamin K2 insufficiency and a high level of dp-ucMGP is a risk factor for death in this population [18]. Vitamin K2 is also a cofactor for the carboxylation of osteocalcin (OC), a protein actively involved in bone remodeling [6]. The highly important role played by vitamin K2 in OC activation is evidenced by an increased risk of bone fractures in case of deficiency of this vitamin [19,20]. There are observations among different populations, including CKD patients, that diet rich in vitamin K, mainly K2, may slow the progression of atherosclerosis and vascular calcification and thereby reduce the risk of CV complications [21,22,23,24,25]. There are a growing number of studies showing that patients with CKD, including KTx, are at risk of vitamin K deficiency [17,18,26], which could be the result, among others, of a restricted diet before the transplantation as well as poor dietary habits after KTx [7]. The current adequate intakes (AIs) for vitamin K for the healthy population are based on the median PK intakes, and for the adult they are: ≥19 y females, 90 μg/day; and ≥19 y males, 120 μg/day [27]. Despite the important role of vitamin K2 in many physiological processes there is no separate dietary requirement for menaquinones [28]. There is lack of special recommendations for daily intake of those vitamins for KTx recipients. The aim of this study was to evaluate the vitamin K1 and K2 content in the diet of patients in the long term after KTx.

## 2. Materials and Methods

Adult patients > 12 months after KTx and under the care of one transplant outpatient clinic in central Poland were considered eligible for the study. Patients using dietary supplements (including any form of vitamin K), with currently diagnosed cancer, symptoms of inflammation (acute or chronic), liver disorders, severe heart failure or advanced graft disfunction (estimated glomerular filtration rate (eGFR) < 15 mL/min/1.73 m^2^), as well as patients following definite diets (vegetarian, vegan, fruit, etc.), were not eligible for the study.

During a routine outpatient transplantation visit, the patients, in the presence of a clinical dietitian, completed a food-frequency questionnaire (FFQ) (Appendix A), a meal diary of our own design. The questionnaire consisted of two parts:I.A complete menu submitted by the patient from the three consecutive working days immediately prior to the visit. The questions concerned the intake and portion sizes of three main meals and any snacks.II.Product groups to accurately determine the food consumed and to obtain information on vitamin K1 and K2 content. The diary listed eight product groups: 1: meats, eggs, fish; 2: dairy products; 3: bread, cereals, pasta; 4: fats; 5: vegetables; 6: fruits; 7: nuts, seeds; 8: sweets.

Using the information from both parts of the diary, a summary of the nutrient composition (including PK and MK) of the consumed food was made using the USDA Food Composition Databases [29]. The MK content of foods was obtained from published sources [30,31,32].

On the same day, the patient’s anthropometric measurements were taken. Height measurement was assessed in the Frankfurt position using a height gauge attached to the medical balance with accuracy of 0.5 cm. Body weight and body mass index (BMI) were determined using a body composition analyzer (Professional Body Composition monitor TANITA BC-545N). Data about medications (including immunosuppressive drugs) and comorbidities such as diabetes mellitus (DM), bone fractures and CV diseases (defined as a history of stroke, myocardial infarction or coronary artery diseases and peripheral atherosclerosis) were obtained from the patients’ medical histories and analysis of their medical records. The graft function was determined by eGFR according to the CKD-EPI (Chronic Kidney Disease–Epidemiology Collaboration) formula on the basis of serum creatinine concentration assessed during the same visit in routine laboratory tests. The results of serum lipid concentration—total cholesterol (TC), HDL cholesterol and triglycerides (TG)—assessed within the previous 6 months were also taken from the medical records. The LDL cholesterol fraction was calculated using the Friedewald formula [33]. TC and TG serum concentrations were considered normal when <200 mg/dL and <150 mg/dL, respectively.

All patients gave informed consent to participate in the study. The study was approved by the Bioethics Committee of the Polish Mother’s Memorial Hospital Research Institute (Decision No 77/2018).

For statistical analysis, the mean, standard deviation, median, minimum, maximum and quartile range were used to analyze the data on dietary intake and food composition. Examination of the distribution of variables was performed using the Shapiro–Wilk test. The nonparametric Mann–Whitney–Wilcoxon test of one sample and for two independent variables and the Kruskal–Wallis test for three or more independent variables were used to compare the data. The nonparametric Wilcoxon test was used as a post-hoc test for significant comparisons with the Kruskal–Wallis test. Spearman’s test was used to observe the correlation of nonparametric variables. Statistical significance was taken as *p* < 0.05. Analyses were performed in the R environment using the following packages: data.table, rstatix, tidyverse, ggpubr, Hmisc, corrplot, PerformanceAnalytics, scales, and gtable.

## 3. Results

A total of 154 KTx patients completed the survey. Three patients were excluded because their eGFR was below 15 mL/min/1.73 m^2^, meaning that ultimately 151 patients, 91 men and 60 women, were eligible for the analysis. The mean age of the KTx respondents was 54.4 ± 12.9 years, with mean 9.7 ± 5.6 years after KTx and with the mean eGFR 53.02 ± 18.7 mL/min/1.73 m^2^. Eight patients had eGFR ≥ 90 mL/min/1.73 m^2^; 39 recipients 60–89.99 mL/min/1.73 m^2^; most of the recipients (*n* = 86) 30–59.99 mL/min/1.73 m^2^; and 18 recipients 15–29.99 mL/min/1.73 m^2^. The mean BMI was 26.4 ± 0.6 kg/m^2^ (18.7–40.8). The patients’ medical statuses were as follows: six patients were after second KTx; all patients had received a kidney from a deceased donor; six patients had a history of CV events, four with myocardial infarction and two with stroke after KTx; 36 recipients had DM, of whom 30 had been diagnosed with post-transplant diabetes (PTDM); 54 KTx patients had bone fractures in their medical history, of which 11 fractures occurred after KTx. Regarding medications, 85 patients were being treated with statins: 70 with atorvastatin, one with lovastatin, eight with rosuvastatin, and six with simvastatin. The mean serum TC was 202.6 ± 43.7 mg/dL, LDL 119.6 ± 34.9 mg/dL, HDL 56.8 ± 13.7 mg/dL, and TG 152.1 ± 70.8 mg/dL. All the patients were treated with antihypertensive medication. Moreover, seven patients were taking vitamin K antagonist (VKA).

Based on the dietary intake analysis, the mean daily intake of vitamin K1 was 120.9 ± 49 μg/day (33.3–436.2 μg/day), whereas that of vitamin K2 was 28.69 ± 11.36 μg/day (2.73–72.21). The intake of particular isoforms of MK are presented in Table 1. The average daily intake of vitamin K1 per kg of body weight was 1.6 ± 0.6 (0.4–4.2) μg/kg, and that of vitamin K2 was 0.4 ± 0.1 μg/kg (0.03–0.9). On the basis of the EFSA recommendation, a deficient intake of vitamin K1 was found in 21 patients (13.9%). The main source of vitamin K1 in our recipients was vegetables (74.1 ± 45 μg), while meat, fish and eggs were the main sources of vitamin K2 (20.3 ± 8.9 μg); see Table 1.

There was no significant difference in vitamin K1 and K2 content in the diets of patients with respect to their graft function (Table 2), except for menaquinone MK-10 in patients with eGFR > 60 mL/min/1.73 m^2^.

There was a significantly higher dietary intake of vitamin K2 in patients with a normal TC level than with hypercholesterolemia (31.2 *±* 10.9 vs. 26.2 *±* 11.3 μg/day; *p* = 0.0037), in particular MK-4 (28.3 ± 9.8 vs. 23.6 ± 9.4; *p* = 0.0038). However, no significant differences were observed in association with TG level (Table 2). A negative correlation was also observed between daily intakes of total vitamin K2 (rho = −0.19, *p* = 0.02), particularly its isoformsMK-4, MK-7 and MK-8, and TC level. A negative association was foundbetween vitamin K2 (rho = −0.22, *p* = 0.006) and MK-4 (rho = −0.2, 0.01) intake and HDL level. There were no significant correlations between the intake of vitamins K1 and K2 and the serum LDL fraction or TG. All correlations between vitamins K daily intakes and lipid levels are presented in Table 3.

There was a significantly higher mean daily intake of vitamin K2 in patients treated with statins compared with those who were not receiving them (30.3 ± 11.12 vs. 6.6 ± 11.5; *p* = 0.028; Table 2).

KTx patients with higher BMIs consumed more vitamin K2 (rho = 0.27, *p* < 0.0001), including MK-4 (rho = 0.25, *p* = 0.001) and MK-10 (rho = 0.16, *p* = 0.04; Table 3). There was no significant difference in total vitamin K1 and K2 intake among patients with or without bone fractures. However, it is noteworthy that patients with bone fractures consumed more MK-10 (0.27 ± 0.6 vs. 0.18 ± 0.5, *p* = 0.011). On the other hand, patients who did not develop bone fractures consumed more MK-5 (0.06 ± 0.1 vs. 0.13 ± 0.2, *p* = 0.018). There was also no relationship observed between the intake of vitamins K1 and K2 in patients with CV episodes; however, this group was very small. Similarly, no significant difference was found in the dietary intake of vitamin K between patients treated and those untreated with VKA, however again, the size of the group taking VKA was very small (*n* = 7). There were no significant differences in vitamin K1 and K2 intake between patients with and without DM. Detailed data are presented in Table 2.

## 4. Discussion

Vitamin K intake varies worldwide and appears to be dependent on dietary habits and varying amounts of vegetables, dairy, meat and fermented foods consumed. Recommendations relating to daily vitamin K intake vary and are not clear. The European Food Safety Authority (EFSA) recommends an Adequate Intake (AI) of 1 µg/kg/day of PK for all age groups except the neonatal period [34]. EFSA recommends an AI of 70 µg/day PK in the diet for a healthy adult population ≥ 18 years for both sexes [34], while the Food and Nutrition Board for the Institute of Medicine recommends a vitamin K1 AI of 120 µg/day for men ≥ 19 years of age and 90 µg/day for women, with no intake recommendation for MK [27]. There is a lack of special dietary recommendations for vitamin K2 intake.

Beulens et al., over a 10-year follow-up of Dutch men and women, showed a daily vitamin K1 intake of 200 ± 98 µg and a daily K2 intake of 31 ± 7 µg [35]. Vitamin K intake in the US adult population is 129.8 ± 8.47 µg/day [36]. High intakes of vitamin K have been observed in China, where men consume 242 µg/day and women 239 µg/day [37]. Nagaoka et al. investigated the diet of KTx patients in Japan and reported a total vitamin K intake of 299 ± 196 µg/day [38], while our study shows that KTx patients consume the amounts of vitamin K1 recommended for the general population but insufficient amounts of vitamin K2. It should be emphasized that, so far, there are no recommendations for vitamin K1 and K2 consumption in KTx populations; however, it is recognized that a diet rich in these vitamins may slow down the progression of vascular and bone abnormalities [5,6,7]. These observations should also draw our attention to the diet of the KTx population. There are many studies showing that KTx recipients are at risk of vitamin K deficiency [17,18] which may be the result not only of maintaining dietary habits from the pretransplantation period but also of dysbacteriosis accompanying uremic toxemia [39] and commonly used medications, including proton pump inhibitors and antibiotics [40]. We studied the consumption of particular forms of vitamin K2, including MK-7, which is considered to be crucial for the prevention of atherosclerosis and vascular calcification, bone fractures and decreasing mortality both in CKD and in the general population [6,16,17,18,19,20,21,22]. Kaneki et al. showed that the consumption of “natto” at least twice a week by Japanese women from Tokyo resulted in high plasma MK-7 concentrations and significantly reduced the risk of bone fractures compared to the Hiroshima region, where “natto” is consumed less than once a week and where more fractures have been reported [41]. In our study population, vegetables were the main source of vitamin K1 (74.1 ± 45 μg), while meat, fish and eggs were the main source of vitamin K2 (20.3 ± 8.9 μg). The intake of vitamin K2 in our study group averaged 28.69 ± 11.36 µg/day, while an intake of 45 µg/day for the adult population is considered to have significant health advantages [42,43,44]. We noticed, furthermore, that patients who had reported CV complications (e.g., heart infarction, brain stroke) had a lower vitamin K1 intake than other patients, but we did not observe such a relationship with vitamin K2 intake (the number of patients with CV complication was very small, though). We also compared the vitamin K2 content of a diet with high lipid levels and observed a negative correlation between vitamin K2 intake and TC, as well as HDL levels, while no such relationship was observed with LDL and TG levels. The study by Beulens et al. showed a higher intake of vitamin K2 with higher HDL concentration in the Dutch population [37]. Braam et al., while analyzing vitamin K1 intake in a healthy lifestyle assessment in the general population, showed an association between higher PK intake an lower TG levels [45]. Our study presents a significantly higher intake of vitamin K1 and K2 in patients treated with statins and in patients with higher BMI, which in the case of vitamin K2 may be related to the higher calorie content of foods containing this form of the vitamin (fatty dairy, meat).

Beulens et al. also showed that higher dietary intake of both forms of vitamin K may reduce the risk of DM in the healthy population [35]. The protective influence may be an effect of stimulation of beta cells of the pancreas to produce insulin as well as of adipocytes to release adiponectin by active forms of MGP and OC, both of which are activated in the presence of vitamin K [46]. In our patients with diabetes, we did not observe differences in dietary intake of vitamin K1 compared with KTx recipients free of this complication; however, they did consume more MK-4.

Multiple studies conducted in the general population show that a high intake of vitamin K1 and K2 has a protective effect on the skeletal system [19,20,47,48,49,50,51]. Knapen et al. emphasized the positive effect of vitamin K2 (MK-4) on the skeletal system, and in a study of 325 postmenopausal women demonstrated that the administration of MK-4 at doses 45 µg/day for 3 years improved the bone mineral content of the femoral neck [51]. Bone disturbances are a frequent complication before and after KTx [2]. Patel et al. in a single-center observational study analyzed bone disorders in a group of 165 KTx patients, and found that up to 44% of them had signs of osteoporosis, while 16% of them had vertebral deformities and some reported low-energy fractures [52]. A total of 54 of our KTx respondents had bone fractures in their medical history and only 11 of them had fractures appeared after KTx. This is too small a number to draw proper conclusions about the relationship between vitamin K intake and the risk of bone fractures; this is one of the limits of our study. Other limitations include an insufficient number of patients compared to large population studies and inadequate group sizes of patients with a particular condition (with DM, CVD etc.). Moreover, our study is cross-sectional, assessing vitamin K intake at only a single timepoint. We additionally failed to assess objective markers of vitamin K1 and K2 deficiency, of bone density and of vascular calcification/atherosclerosis. However, considering the role that both vitamins K are presumed to play in various important processes, mainly in cardiovascular and bone health, it seems advisable to pay attention to their content in the diet. Despite the existing European supplementation recommendations for vitamin K1, there are still no established standards for dietary vitamin K intake and supplementation in KTx patients. Therefore, further studies are needed on the dietary recommendations for both forms of vitamin K in patients with CKD, especially after KTx.

## 5. Conclusions

The daily intake of vitamin K1 in KTx patients living in Poland is adequate, but their diet appears to be poor in vitamin K2. Further studies are needed in order to evaluate proper recommendations for the daily vitamin K1 and K2 intake of KTx recipients, as well as the necessity of its possible supplementation.

## Figures and Tables

**Table 1 nutrients-14-05070-t001:** The intake of vitamins K1 and K2 and its isoforms: MK4–MK10 in consumed product groups by kidney transplant patients.

	Average Daily Total Intake (μg) (Mean ± SD)								
	Total	Meat, Eggs, Fish	Seeds and Nuts	Fruits	Bread, Cereals, Pasta	Dairy Products	Sweets	Fats	Vegetables
K1	120.9 ± 49	1.5 ± 1.7	0.02 ± 0.2	8.7 ± 9.4	6.2 ± 3.2	1 ± 0.7	17.7 ± 21.9	11.5 ± 9.8	74.1 ± 45
K2	28.7 ± 11.3	20.3 ± 8.9	-	-	-	4.7 ± 6.5	0.9 ± 4.3	1.8 ± 1.2	1.7 ± 4.6
K2: MK-4	25.9 ± 9.9	19.6 ± 8.9				3.1 ± 2.9	0.04 ± 0.07	0.01 ± 0.05	0.04 ± 0.2
K2: MK-5	0.11 ± 0.2	-				0.9 ± 4.3	-	-	0.0003 ± 0.002
K2: MK-6	0.24 ± 0.4	0.1 ± 0.1				1.8 ± 1.2	-	-	-
K2: MK-7	0.25 ± 0.27	0.2 ± 0.2				1.3 ± 4.4	0.07 ± 0.2	0.13 ± 0.3	0.02 ± 0.04
K2: MK-8	1 ± 1.9	0.4 ± 0.4				3.1 ± 2.9	0.04 ± 0.07	0.01 ± 0.05	0.04 ± 0.2
K2: MK-9	0.9 ± 2.3	-				0.9 ± 4.3	-	-	0.0003 ± 0.002
K2: MK-10	0.2 ± 0.5	-				1.8 ± 1.2	-	-	-

**Table 2 nutrients-14-05070-t002:** The daily intake of vitamins K1, K2 and its isoforms MK-4 to MK-10 in relation to comorbidities, medications taken and selected laboratory parameters.

	K1 (μg/d) ± SD	K2 (μg/d) ± SD	Menaquinones						
			MK-4 (μg/d) ± SD	MK-5 (μg/d) ± SD	MK-6 (μg/d) ± SD	MK-7 (μg/d) ± SD	MK-8 (μg/d) ± SD	MK-9 (μg/d) ± SD	MK-10 (μg/d) ± SD
Cardiovascular incidents									
-yes	92.4 ± 32.9	26.6 ± 11	25.2 ± 11.3	0.06 ± 0.1	0.23 ± 0.4	0.22 ± 0.2	0.5 ± 0.5	0.2 ± 0.2	0.15 ± 0.1
-no	122 ± 49.3	28.8 ± 11.4	25.9 ± 9.8	0.1 ± 0.01	0.26 ± 0.4	0.26 ± 0.3	1.1 ± 1.9	0.9 ± 2.4	0.2 ± 0.5
*p*-value	0.089	0.61	0.73	0.34	0.87	0.54	0.55	0.76	0.39
Bone fractures									
-yes	124.8 ± 56.8	26.7 ± 11	24.6 ± 10.6	0.06 ± 0.1	0.18 ± 0.3	0.24 ± 0.3	0.7 ± 1	0.5 ± 1.4	0.27 ± 0.6
-no	118.3 ± 44.3	29.8 ± 11.5	26.6 ± 9.4	0.13 ± 0.2	0.27 ± 0.4	0.26 ± 0.3	1.2 ± 2.2	1.2 ± 2.7	0.18 ± 0.5
*p*-value	0.55	0.18	0.21	0.018	0.21	0.34	0.28	0.28	0.011
Statins									
-yes	120.7 ± 57.4	30.3 ± 11.12	27.1 ± 9	0.1 ± 0.2	0.26 ± 0.4	0.27 ± 0.3	1.2 ± 2	1.1 ± 2.6	0.22 ± 0.6
-no	121 ± 35.8	6.6 ± 11.5	24.4 ± 10.7	0.1 ± 0.2	0.21 ± 0.3	0.23 ± 0.3	0.9 ± 1.7	0.65 ± 1.9	0.2 ± 0.5
*p*-value	0.37	0.028	0.032	0.82	0.47	0.22	0.2	0.26	0.6
DM									
-yes	121.9 ± 38.6	31 ± 12.2	27.8 ± 8	0.08 ± 0.1	0.18 ± 0.3	0.25 ± 0.2	1.29 ± 2.6	1.1 ± 3.2	0.19 ± 0.4
-no	120.5 ± 52	28 ± 11	25.3 ± 10.3	0.1 ± 0.2	0.26 ± 0.4	0.26 ± 0.3	0.98 ± 1.6	0.8 ± 1.9	0.2 ± 0.6
*p*-value	0.67	0.18	0.064	0.29	0.37	0.99	0.91	0.59	0.97
eGFR									
mL/min/1.73 m^2^									
≥30	121.8 ± 49.9	29 ± 11.7	26.2 ± 10.1	0.11 ± 0.19	0.24 ± 0.03	0.25 ± 0.26	1.1 ± 2	0.93 ± 2.42	0.19 ± 0.46
<30	113.8 ± 42.5	26 ± 8.2	23.7 ± 7.5	0.08 ± 0.14	0.26 ± 0.05	0.3 ± 0.34	0.67 ± 0.6	0.61 ± 1.4	0.34 ± 0.96
*p*-value	0.66	0.29	0.29	0.41	0.19	0.68	0.88	0.77	0.78
≥45	122.8 ± 51.4	28.8 ± 11.4	26.1 ± 10	0.1 ± 0.18	0.24 ± 0.36	0.23 ± 0.25	1.1 ± 1.98	0.89 ± 2.32	0.17 ± 0.46
<45	117 ± 44.1	28.4 ± 11.4	25.5 ± 9.6	0.12 ± 0.18	0.24 ± 0.36	0.3 ± 0.3	0.98 ± 1.71	0.89 ± 2.35	0.29 ± 0.68
*p*-value	0.85	0.57	0.49	0.36	0.68	0.2	0.43	0.1	0.097
≥60	122.8 ± 43.9	28.7 ± 12.2	25.6 ± 9.7	0.12 ± 0.2	0.27 ± 0.4	0.2 ± 0.2	1.3 ± 2.6	1.1 ± 2.9	0.07 ± 0.1
<60	120 ± 51.3	28.7 ± 11	26.1 ± 10	0.1 ± 0.2	0.23 ± 0.3	0.28 ± 0.3	0.9 ± 1.5	0.8 ± 2	0.3 ± 0.6
*p*-value	0.84	0.71	0.65	0.54	0.6	0.13	0.46	0.12	0.013
Total cholesterol (TC)									
<200 mg/dL	126 ± 59.3	31.2 ± 10.9	28.3 ± 9.8	0.09 ± 0.2	0.2 ± 0.4	0.3 ± 0.3	1.1 ± 1.8	0.9 ± 2.2	0.2 ± 0.5
≥200 mg/dL	115.8 ± 35.8	26.2 ± 11.3	23.6 ± 9.4	0.1 ± 0.2	0.2 ± 0.3	0.2 ± 0.3	1 ± 2	0.9 ± 2.4	0.2 ± 0.5
*p*-value	0.61	0.0037	0.0038	0.13	0.95	0.12	0.071	0.33	0.05
Triglycerides (TG)									
<150 mg/dL	116 ± 39.5	28 ± 11.1	25.6 ± 10.1	0.1 ± 0.2	0.3 ± 0.4	0.3 ± 0.3	0.9 ± 1.4	0.6 ± 1.8	0.2 ± 0.6
≥150 mg/dL	127.3 ± 59.2	29.7 ± 11.7	26.4 ± 9.6	0.1 ± 0.2	0.2 ± 0.3	0.2 ± 0.2	1.3 ± 2.3	1.2 ± 2.9	0.2 ± 0.4
*p*-value	0.43	0.35	0.54	0.85	0.77	0.52	0.83	0.76	0.58

**Table 3 nutrients-14-05070-t003:** Correlation of daily intake of vitamins K1 and K2 and its isoforms MK-4 to MK-10 in relation to selected anthropometric and laboratory parameters.

	K1	K2	Isoform K2						
			MK-4	MK-5	MK-6	MK-7	MK-8	MK-9	MK-10
BMI (kg/m^2^)									
rho	0.02	0.27	0.25	−0.08	−0.11	0.05	0.09	0.14	0.16
*p*-value	0.8	<0.0001	0.001	0.32	0.17	0.5	0.26	0.07	0.04
Total cholesterol (TC)									
rho	−0.08	−0.19	−0.18	0.11	−0.01	−0.17	−0.17	−0.12	−0.2
*p*-value	0.34	0.02	0.03	0.19	0.86	0.04	0.04	0.15	0.15
LDL									
rho	−0.07	−0.12	−0.12	0.1	−0.05	−0.14	−0.09	−0.06	−0.14
*p*-value	0.38	0.13	0.14	0.2	0.52	0.1	0.26	0.44	0.08
HDL									
rho	−0.11	−0.22	−0.2	0.07	0.07	−0.07	−0.15	−0.11	−0.1
*p*-value	0.17	0.006	0.01	0.37	0.39	0.37	0.07	0.18	0.21
Triglycerides									
rho	0.06	0.09	0.05	−0.07	−0.04	−0.1	0.003	0.04	−0.02
*p*-value	0.5	0.25	0.51	0.4	0.65	0.24	0.97	0.65	0.8

## Data Availability

The data presented in this study are available on request from the corresponding author. The data are not publicly available due to being clinical data.

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
