# Peer review of "Vitamin K1 and K2 in the Diet of Patients in the Long Term after Kidney Transplantation"

_nutrients, 2022, doi:10.3390/nu14235070_

Round 1
Reviewer 1 Report
Comments of this reviewer are:
There are many limitations, mainly the lack of measurement of objective markers of vitamin K1 and K2 deficiency, that the authors have already acknowledged.
Author Response
There are many limitations, mainly the lack of measurement of objective markers of vitamin K1 and K2 deficiency, that the authors have already acknowledged.
Thank you very much for your review. We agree that the lack of objective markers of vitamin K1 and K2 deficiency are the main limitation of our study. We are planning to perform such measurements in another study.

Reviewer 2 Report
Kurnatowska and coworkers presented an interesting study: Vitamin K1 and K2 in the diet of patients in long time after kidney transplantation.
The authors addressed an interested topic, enrolled more than 150 kidney transplant patients, used appropriate methodology, highlighted the strengths and limitation of their study, and presented some interesting results and conclusions.
However, closer look raised some comments:
1. It would be interesting to present a meal diary as an appendix.
2. Is it possible to predict the patient's eating habits based on the menu for the three working days before the outpatient clinic visit?
3. The flowchart of the study design would be of benefit.
4. Did the authors check the proton pump inhibitors use in their patients?
5. Can the authors make any practical dietary recommendations for patients after kidney transplantation?
6. The explanation of the MK abbreviation in the abstract is missing.
The authors should accept and discuss these comments.
Author Response
Comments and Suggestions for Authors
Kurnatowska and coworkers presented an interesting study: Vitamin K1 and K2 in the diet of patients in long time after kidney transplantation.
The authors addressed an interested topic, enrolled more than 150 kidney transplant patients, used appropriate methodology, highlighted the strengths and limitation of their study, and presented some interesting results and conclusions.
However, closer look raised some comments:
- It would be interesting to present a meal diary as an appendix.
In the attachment please find translated Meal Diary as well as Polish version of photo album of food and meals given for the respondent’s use.
- Is it possible to predict the patient’s eating habits based on the menu for the three working days before the outpatient clinic visit?
As a standard was taken a meal consumption for three working days in a row. It is the most objective and reliable as well as commonly used research way. Patient is able to credibly present his daily menu and remembers what he/she ate in the last 3 days. It allows to avoid mistakes and helps to recollect if particular products from the diary were eaten. Also the attachment of a photo album of products is a very useful tool in order to determine mass/weight of eaten products. The visualization helps patient to show precise amount of food consumed.
- The flowchart of the study design would be of benefit.
Thank you very much for the suggestion. Our study is based on conducting a one-time nutrition survey.
The scheme of the research was described in “Material and methods” quite precisely and was not complicated. It seems that creating a flowchart for this type of the research is pretty difficult to manage and not so advisable because all the participating respondents gave consent and fulfilled inclusion/exclusion criteria.
- Did the authors check the proton pump inhibitors use in their patients?
It is a valuable suggestion to check the IPP intake however we did not do it in our study. However it seems that IPP use does not have influence on patient diet and very often it is used without clinical indications (in Poland IPP are OTC drugs). Nevertheless the use of IPP may have an impact on absorption also of vitamin K, yet in our research we did not check vitamin K concentrations which is undoubtedly a limitation of our study (mentioned in the text). However we believe it should not have impact on patients diet.
- Can the authors make any practical dietary recommendations for patients after kidney transplantation?
We did not give any dietary recommendations for kidney transplant recipients because it was not a purpose of our study and it is a topic for a different research. Dietary recommendations for KTx are dependent on many different factors such as time after transplantation, graft function, used immunosuppression drugs and comorbidities. In general after long period from KTx the diet should be healthy, with calories limitations up to 30-35 kg per weight/per day, with salt intake restriction up to <2-3 grams per day and limitation of sugar. We have sent for this journal issue another article about diet in kidney transplant recipients: Górska M, Kurnatowska I: Nutrition disturbances and metabolic complications in kidney
transplant recipients: etiology, methods of assessment and prevention - review.
- The explanation of the MK abbreviation in the abstract is missing.
Thank you, the explanation has been added.
